# Exposure to asylum seekers and changing support for the radical right: A natural experiment in the Netherlands

**Jochem Tolsma◉\*, Joran Laméris, Michael Savelkoul**

Sociology, Radboud University, Nijmegen, The Netherlands

\* j.tolsma@ru.nl

**Data Availability Statement:** All relevant data are within the manuscript, its Supporting Information files and its replication package at GitHub (https://jochemtolsma.github.io/ExposureASC2020/).

## Abstract

As a result of the 2015 refugee crisis, a substantial number of voters experienced a sudden and unexpected influx of asylum seekers in their neighbourhood in the Netherlands. We examined whether and why local exposure to asylum seekers leads to more support for the radical right (i.e. PVV). Our analyses are based on a longitudinal individual-level panel data-set including more than 19,000 respondents (1VOP) who were interviewed just before and shortly after the height of the refugee crisis. We enriched this dataset with detailed information about where asylum seekers were housed from the Central Agency for the Reception of Asylum Seekers. Our empirical study resembles a natural experiment, because some residents experienced an increase in exposure to asylum seekers but similar residents did not. PVV support increased during the refugee crisis and especially among residents who became more exposed to asylum seekers in their neighbourhood.

## Introduction

An unprecedented refugee crisis unfolded in Europe over the course of 2015, which brought about political turmoil in many countries. In this study we take a closer look at changes in support for the radical right in the Netherlands against the background of a large influx of asylum seekers in that year. As some people witnessed an influx of asylum seekers in their residential neighbourhood, while others did not, the 2015 European refugee crisis is an interesting case to study the impact of local demographic changes on voting intensions for the radical right.

In 2015, the twenty-eight member states of the European Union together with Norway and Switzerland received more than 1.3 million applications from asylum seekers [1]. This has been, by far, the highest annual number of asylum seekers recorded in Europe since World War II [2]. The majority of the asylum seekers that arrived in Europe in 2015 came from Muslim-majority countries. In the Netherlands–the current site of study–more than 40,000 asylum seekers (255 applicants per 100,000 inhabitants) entered the country in 2015, a historical record. The centre-left (PvdA) and centre-right party (VVD) forming the Dutch government fiercely debated the development of a strategy for dealing with these asylum seekers. The refugee crisis not only evoked divergent reactions among Dutch politicians but also among the broader public. The Dutch people made their voice heard, both in support of and in opposition

**Funding:** The authors received no specific funding for this work.

**Competing interests:** The authors have declared that no competing interests exist.

to the arrival of asylum seekers. While the Netherlands Red Cross saw their stock of temporary volunteers grow from 6,000 to more than 36,000 [3], public demonstrations against the arrival of asylum seekers also intensified over the course of 2015. Especially in communities where new asylum seeker centres (ASCs) were established, people uttered their disapproval by, for example, hanging banners on the assigned buildings with anti-refugee slogans, such as 'Own people first'.

In this study, we take advantage of a large-scale longitudinal individual-level panel dataset to examine changes in support for the radical right against the background of the sudden arrival of asylum seekers in the Netherlands. In particular, we are interested in whether voters changed their intentions to vote for the radical right after having experienced an inflow of asylum seekers in their residential neighbourhood.

The *Partij voor de Vrijheid* (Party for Freedom; PVV) was in 2015 the only radical right party in the Netherlands with seats in Parliament [4,5]. The PVV was founded in 2006 and since then led by Geert Wilders. The PVV is known for its anti-immigration statements and its campaign to 'de-Islamize' the country. During the refugee crisis, the PVV started targeting (Muslim) asylum seekers in its political discourse. Wilders announced in October 2015 the launch of a website where people can report complaints about asylum seekers [6]. During new year's eve 2015, mass sexual assaults took place in Cologne, Germany. Relatively many of the identified suspects were asylum seekers [7]. Wilders subsequently propagated that all male asylum seekers should be locked up [8]. The PVV is, therefore, an attractive party for voters with (strong) anti-migration, anti-Muslim and anti-asylum seeker attitudes.

The ethnic composition of people's living environment is a focal point in scholarly attempts to explain the popularity of radical right parties [9–12]. In the Netherlands, support for the radical right among native-Dutch is more common in neighbourhoods where more non-western migrants (and their descendants) live [13,14]. Yet, war refugees are clearly a distinct group compared to immigrants who enter destination countries such as the Netherlands as a result of union formation, family reunification and labour migration. It is likely that, in general, war refugees are considered to be more deserving to enter and stay in receiving countries [15] and that they may therefore provoke less opposition.

Having said that, previous research has shown that a larger influx of asylum seekers is positively related with more support for the radical right [16,17] (but see also [18]). However, studies addressing this relationship are largely restricted to the country-level and generally draw on cross-sectional data collected before the 2015 European refugee crisis. Research addressing the influx of asylum seekers in local residential areas and its consequences for people's voting intentions is scarce. Dustmann, Vasiljeva and Damm [19] showed that the allocation of larger refugee shares in Danish municipalities in the 1980's and 1990's was related with higher vote shares for anti-immigration parties. Dinas and colleagues [20] demonstrated that the massive but transient inflow of refugees on Aegean islands during the 2015 European refugee crisis fuelled support for the extreme-right on these islands. Besides focusing on relatively large geographical units, both studies employed a macro-level approach–i.e., focusing on vote shares, rather than individual voting intensions and changes thereof over time–which does not allow to empirically assess underlying mechanisms for this relationship.

In this study, we aim to contribute to earlier research in two ways. Taking advantage of individual-level longitudinal panel data, we will first examine whether Dutch voters who have been exposed to a sudden and unexpected influx of asylum seekers in their local neighbourhood are more (or less) likely to *change* their voting intention to the PVV than voters who have not been exposed to an inflow of asylum seekers in their neighbourhood. Second, we aim to explain any observed relationship between the influx of asylum seekers in the neighbourhood and support for the radical right.

A positive relationship between the two phenomena may be explained by the threat mechanism [21–23] stating that an increasing ethnic outgroup size fosters feelings of economic and cultural ethnic group threat, and consequently, anti-immigration attitudes which are the central attitudinal driving force behind support for the radical right [24,25]. At the same time, residential proximity to ASCs may lead to contact with asylum seekers [26]. Positive intergroup contact stimulates interethnic tolerance [27,28], which can be linked to lower levels of support for the radical right. The few, cross-sectional studies putting both mechanisms to the test, found support for both the threat and the positive contact mechanism, albeit weaker for the latter [11,14,29]. We are the first to test both the threat and positive contact mechanisms simultaneously from a longitudinal perspective, using individual-level panel data in the context of the refugee crisis.

To reach these aims, we employ a longitudinal and sizeable panel dataset on individual respondents (N>19,000; 1Vandaag Opinion Panel Survey). Our panel dataset allows us to control for (time-stable) unobserved heterogeneity. We enriched this micro-level data with detailed information about where asylum seekers were housed from the Central Agency for the Reception of Asylum Seekers (COA). Formally, the term 'asylum seeker' refers to persons who apply for asylum and seek refugee status. We therefore use this term when we talk about the persons who were housed in the ASCs during the 2015 refugee crisis. We use the term 'refugee' more loosely to refer to persons fleeing the risk of serious harm and persecution (and who may or may not have an official refugee status).

The period in between the two waves of our data (February and November 2015) spanned the period in which the settlement of asylum seekers in residential environments throughout the Netherlands took place. Asylum seekers were unable to select the region where they wanted to be housed. Similarly, neighbourhood residents had no, or only limited influence, in where new asylum seekers were going to be housed and, within the time-window of our study, did not have time to move out of their neighbourhood if they opposed to the inflow of asylum seekers. Selective residential mobility, generally plaguing the neighbourhood effects literature, is thus not an issue. Given the sudden and unexpected influx of asylum seekers, the management of asylum seekers flows was chaotic and haphazard. The placement of asylum seekers was–as we will show below–to a large extent random and therefore our study resembles to some extent a natural experiment. For these three reasons–individual-level panel data allowing to control for (time-stable) unobserved heterogeneity, no selective residential mobility, increased exposure to asylum seekers (to a large extent) random–we are thus able to make relatively strong causal claims on the impact of the refugee crisis on changing support for the PVV.

## Theoretical expectations

According to conflict theories [21–23], a larger relative size of non-natives in people's living environment induces competition between natives and non-natives for scarce economic resources (e.g. jobs and affordable housing). Conflicting cultural values (e.g. toward homosexuals or freedom of speech), also, become more apparent when the group of non-natives is more sizeable. After reviewing approximately 100 studies of immigration attitudes, Hainmueller and Hopkins [30] conclude that perceptions of group threat affect immigration attitudes. This holds especially for concerns about the cultural impact of immigration, and not so much the possible consequences of immigration for one's personal (economic) situation. Because earlier research convincingly showed that voters who perceive ethnic minorities as a threat and who hold anti-immigration attitudes are more likely to cast their vote for the radical right

[11,31,32], one would also expect to observe a positive relationship between the presence of non-natives in people's living environment and their likelihood to vote for the radical right.

Previous empirical studies reached mixed conclusions when it comes to the relationship between the ethnic composition of people's *local* living environment and their likelihood to vote for the radical right. Whereas several studies provided support for a positive relationship [9,12,33], others found no significant relationship [34] or even a negative relationship [35,36]. For the Netherlands, previous studies showed that support for the radical right is more common in neighbourhoods with a larger share of non-western migrants [13,14] (but see also Van Wijk *et al.* [37] who found a U-shaped relationship). These studies lack, however, convincing evidence for a threat mechanism. This was due to the fact that outgroup sizes at the neighbourhood level were not consistently linked to more intense feelings of ethnic group threat, a null-finding reported by others as well [38–40].

Olzak [41] was one of the first scholars who argued that recent substantial increases rather than stable levels of non-natives in people's living environment trigger perceptions of ethnic threat. In line with this idea, several studies provided empirical evidence for a positive relationship between an increase in immigrants and voting for the radical right [17,34] (but see also Lubbers *et al.* [42]). The 2015 European refugee crisis provides an interesting case to test the impact of a sudden demographic change on radical right voting. At the country-level, several studies provided support for a positive relationship between the share as well as the influx of asylum seekers and support for the radical right [16,17] (but see also Arzheimer and Carter [18]). Similar relationships have been found at somewhat lower geographic scales [19,20].

Although asylum seekers in the Netherlands do not directly compete with natives for jobs–they are not allowed to work–they do receive a small allowance and, once they are granted a refugee and permanent resident status, will compete for public housing. This is likely to trigger perceptions of economic threat among natives. Asylum seekers in the Netherlands are not confined to ASCs and are allowed to wander free in the neighbourhood and further. Because of the cultural distance between asylum seekers and natives, natives may perceive more cultural and safety threat as well. Previous research provided tentative support for a positive relationship between the share of asylum seekers and perceptions of ethnic threat as well as negative stances towards immigrants [43,44] (but see also Scheepers *et al.* [45]). We thus expect that voters who have suddenly become exposed to asylum seekers in their neighbourhood are more likely to support the PVV than their counterparts in neighbourhoods who did not experience a sudden influx of asylum seekers (Hypothesis 1). And, on the basis of conflict theory, that increased support for the PVV as a consequence of increased local exposure to asylum seekers may be explained by increased perceptions of local intergroup threat (Hypothesis 2).

Yet, the macro-structural theory of intergroup relations [26] and contact theory [27] would suggest a different causal mechanism. Residential proximity to ASCs may provide the opportunity to interact with asylum seekers [26]. Intergroup contact, when positive, fosters tolerance [27,28]. Hence, positive contact with asylum seekers may make voters less inclined to support the radical right. The influence of positive contact experiences can be explained by induced levels of knowledge, empathy and perspective taking [28,46]. According to Allport,—superficial contact (i.e. mere exposure) or overt negative contact, like abuse and name-calling would only lead to more trouble [27]. Intergroup contact would only foster tolerance if contact takes place under 'optimal' conditions, like equal group status or common objectives. However, Pettigrew and Tropp [28] convincingly demonstrated in their meta-study that although the effect of intergroup contact is stronger if contact takes place under optimal conditions, commonly, these conditions are not necessary; intergroup contact–at least when not clearly negative–induces tolerance even if Allport's conditions are not met (but see also Paluck *et al.* [47] who point at a lack of experimental evidence for this claim).

Based on a choice experiment on preferences for refugee and migrant homes, Liebe et al. [48] conclude that contact with refugees increases acceptance to refugee homes in the immediate vicinity, in line with contact theory. Studies that focused on the impact of intergroup contact on radical right voting seem to provide support for both the positive contact mechanism [11,14,29] (but see also Savelkoul and Scheepers [49]) and the negative contact mechanism (Nijs *et al.* [50]). However, the presence of asylum seekers in ASCs may not always provide the most favourable opportunities for sustained positive interactions [20]. It is thus not self-evidently true that increased proximity to asylum seekers leads to more positive contact experiences [51], or that it not at the same time leads to more negative contact experiences. That being said, we expect, on the basis of contact theory, that the presumed positive relation between increased local exposure to asylum seekers and increased support for the PVV as a result of feelings of ethnic threat may be supressed by increased local positive interethnic contact experiences (Hypothesis 3).

## Data and operationalization

### 1Vandaag Opinion Panel

This study employs individual-level panel data from the 1Vandaag Opinion Panel (1VOP) in the Netherlands. The 1Vandaag Opinion Panel consists of 50,000 people from all parts of the Dutch population living across the country. People sign up for this online panel of their own. Every week panel members give their opinion on current topics such as politics, economics, health care and crime. The results are announced in the broadcasts of 'EenVandaag' on public television and presented to politicians and policymakers. The advisory board of the 1VOP consists of Dutch University professors Joop van Holsteijn, Jelke Bethlehem and Tom van der Meer. More information on the panel can be found here (in Dutch): https://eenvandaag. avrotros.nl/panels/opiniepanel/uitleg/. For access to the original (anonymized) data we received from 1VOP, scholars may contact the owners of the 1VOP. For the current study we did not access any personal identifying data. Our study does not pose any risk to panel members or their individual privacy and hence we did not deem it necessary to seek approval of Radboud University's Ethics Committee.

The measures used in this contribution are included in two waves of the 1VOP. Because people sign up for this online panel of their own, there is a self-selection bias in the sample of respondents [52]. However, due to a uniquely large sample size we cover a high degree of the variety of people found within the Dutch population even though some groups (e.g. women, the young, lower educated) are underrepresented in the sample. The first wave of our data was collected in February 2015 and the second wave of our data was collected in November 2015. The period in between the two waves spanned the period in which the high influx of asylum seekers and the subsequent settlement of asylum seekers in residential environments throughout the Netherlands took place (Fig 1). As the focus of our study is on native Dutch individuals, we excluded respondents with a non-native Dutch background from our analyses (N = 1,151). We classified respondents as native Dutch when both parents were Dutch, or when respondents identified as being Dutch in case one parent was non-Dutch.

1VOP panel members regularly receive invitations to fill in online questionnaires and can decide whether or not to participate in specific waves. Of the 26,064 native Dutch respondents who filled out the questionnaire in wave 1, 19,988 respondents also completed the questionnaire in wave 2. There was no significant relationship between becoming exposed to asylum seekers in the local environment, the main focus of our analysis, and the likelihood of participating in wave 2. PVV supporters of wave 1 had a .79 probability to participate in wave 2, non-PVV supporters a .76 probability. This selectively will lead to conservative tests of our

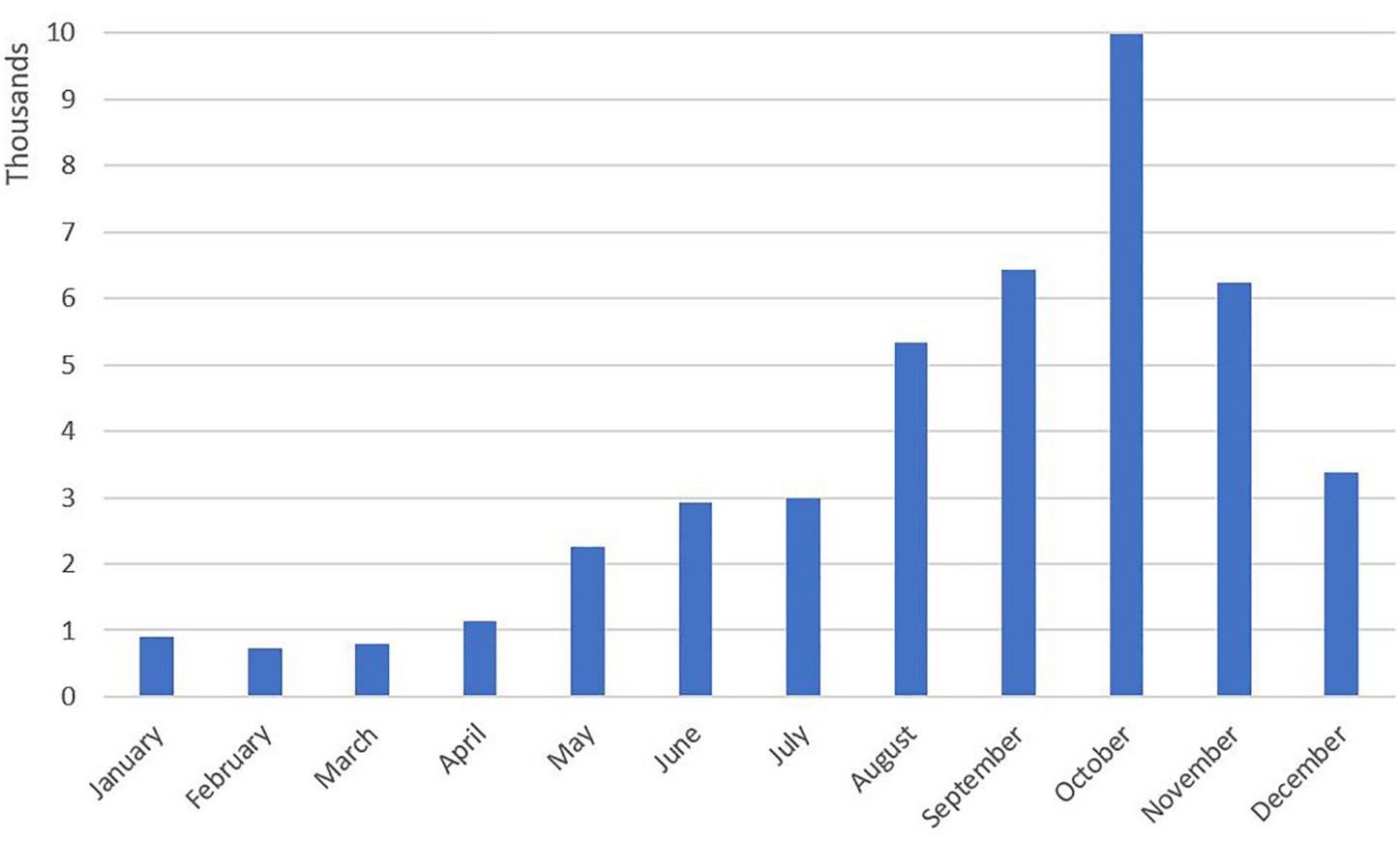

**Fig 1. Number of asylum requests per month in 2015 in the Netherlands.** Source: Statistics Netherlands.

hypotheses; we will be less likely to pick up an increase in PVV support over time in our panel data (see Table A1 in S1 Appendix).

The neighbourhood identifier included in the 1VOP is the four-digit part of the post code. The median number of residents in these neighbourhoods is 2,645 (mean = 4,142) and the median surface area 5.35km$^2$ (mean = 8.65km$^2$).

## Central agency for the reception of asylum seekers

We enriched our individual-level data with information from the Central Agency for the Reception of Asylum Seekers (COA) about the number of asylum seekers at the neighbourhood-level. COA is responsible for housing asylum seekers from the time they request asylum until they receive a residence permit or must leave the Netherlands. Due to the exceptionally high influx of asylum seekers in 2015, the maximum capacity of existing regular reception centres ('reguliere opvang') was soon reached. COA therefore opened new regular reception centres but also housed asylum seekers in temporary centres ('noodopvang') and, starting from September 2015, also in crisis centres ('crisisnoodopvang'). Regular reception centres are used for at least a period of two years and have a capacity ranging from 300 to more than 1,500 people. Temporary centres were set up in, for example, remodelled market halls or empty office buildings. These centres generally house around 300 asylum seekers and are used for a period of six to twelve months. Various facilities were used as crisis centres, such as sport halls and old school buildings, which were already marked out by local governments to house citizens in

times of incidents or disasters. Crisis centres give room to a dozen to several hundred asylum seekers, but only for short periods (in principle up to 72 hours) at a time. Before asylum seekers could be housed in crisis centres by COA, the local government needed–in principle–to agree. However, the placement process was chaotic and even for policymakers the procedures and responsibilities were unclear [53].

Asylum seekers did not have any say in where they were going to be housed and because we assess changes in voting intention in a relatively short period of time, native residents are very unlikely to have moved out of the neighbourhood as a consequence of the inflow of asylum seekers. Selective residential mobility plaguing neighbourhood effects research in general [54] is thus unlikely to influence this study's results.

Notwithstanding that protests of the local population were more intense in some places than in others and that some municipal governments displayed a higher willingness to host asylum seekers than others, the increased exposure to asylum seekers was to a large extent an exogenous process for the neighbourhood residents whose voting intention it could affect. Voters of neighbourhoods that would experience an inflow of asylum seekers did not differ with respect to radical right support from voters that would not experience an inflow of asylum seekers: pre-crisis support for the PVV for both the 'treated' and 'untreated' groups was approximately 17%.

## Changes in support for the PVV

To examine changes in voting intention for the radical right in the Netherlands, we measured respondents' intended voting behaviour at two time points with the following question: 'Which party would you vote for if parliamentary elections were held today?'. The answer categories consisted of the eleven largest political parties represented in the Dutch parliament as well as the option 'another party'. In addition, respondents could also answer 'I don't know', 'blank vote', 'I'm allowed to vote, but I wouldn't', I'm not allowed to vote', and 'no answer'. We removed from the analysis the respondents who answered 'I'm not allowed to vote' and 'no answer' in one or both waves (less than 2%). Respondents who answered 'I don't know', 'blank vote', 'I'm allowed to vote, but I wouldn't' could in one of the two waves be politically mobilized or demobilized and experience respectively a shift towards or a shift away from the radical right and are therefore included in our analysis. Voting intention for the PVV is included as dichotomized variable (YES = 1 vs. NO = 0).

## Exposure to asylum seekers

For each ASC we know the exact address. Depending on the type of ASC we received different information from COA. For regular and temporary ASCs we know how many asylum seekers were housed at 1-1-2015 and 15-11-2015. For each crisis centres we know the daily mutations in asylum seekers from 18-9-2015 until 15-11-2015. Before 18 September, asylum seekers were not housed in crisis centres. The data of COA thus allowed us to capture the change in exposure that took place *after* respondents were interviewed for the first time and *before* respondents were interviewed for the second time. To account for differences in sizes between neighbourhoods, we calculate the number of asylum seekers per 1,000 inhabitants. We acknowledge that the exposure to asylum seekers and the impact of this exposure for voting intentions may depend on the type of ASC asylum seekers were housed in. Native residents living close by regular ASCs may already have been familiarized by the presence of asylum seekers [55]. This may mitigate the impact of increased exposure on PVV support. Asylum seekers could only live very briefly in the same crisis ASCs (generally up to 72 hours), which will have made it difficult to develop sustained positive interactions with native neighbourhood

residents. Especially among neighbourhood residents who have become exposed to asylum seekers as a consequence of placement of asylum seekers in crisis ASCs, we may therefore expect the threat mechanism to dominate. Thus, besides the measure of total relative exposure to asylum seekers, we also calculate the increase in the number of asylum seekers in regular ASCs, in temporary ASCs, and in crisis ASCs separately.

### Ethnic threat and interethnic contact

Feelings of local interethnic threat are measured with the item: 'I sometimes worry about the fact that my neighbourhood deteriorates because of the arrival of ethnic minorities'. The answer categories are: 0. 'totally disagree', 1. 'disagree', 2. 'agree/nor disagree'/'I don't know/no opinion', 3. 'agree', and 4. 'totally agree'. We measure local positive interethnic contact with non-western ethnic minorities with the following question: 'How often do you have personal contact in your neighbourhood with people of non-western descent? By personal contact, we mean that you know the name of this person and occasionally have a conversation with this person.'. The answer categories to this item are: 0.'never'/'not applicable', 1.'about once a year', 2.'several times a year', 3.'about once a month', 4.'several times a month', 5.'several times a week', and 6.'(almost) every day'.

### Missing values and working sample

We removed 506 respondents (2.5%) for whom we could not match the contextual information about the exposure to asylum seekers in the local living environment due to missing information on their geographical location. This left us with a working sample of 19,091 respondents in 2,997 four-digit postcode areas (74% of all inhabited postcode areas). Descriptive statistics of our main variables–based on the final samples used in our analyses–are displayed in Table 1.

## Analytical strategy

To test our hypotheses, we employ both fixed effects and hybrid models. In fixed effects models the influence of all time-invariant characteristics are removed, allowing us to assess the net effect of being exposed (more precisely: changes in exposure) to asylum seekers in the local living environment on individuals' changes in voting intentions. The fixed effects analyses are based on a reduced sample of the respondents whose voting intentions changed over time (N = 1,389, living in 1,002 neighbourhoods). Fixed effects models tell us what would happen to an individual's voting intention if the exposure to asylum seekers would increase by one unit given that these individuals have changed their voting intention between time point 1 and time point 2. Because voting intentions for the PVV are operationalized as a dichotomous variable, we estimate logistic fixed effects models.

In our hybrid models (aka 'between-within method'), time-varying predictors are decomposed into a between-person component (i.e. person-specific mean) and a within-person component (i.e. deviation from person-specific mean). Time (i.e. wave) is included as fixed-effect. The causal estimates tell us how experiencing a change (e.g. in exposure to asylum seekers) is related to a change in the odds to vote for the PVV. An advantage of this method is that respondents who did not experienced a change in voting intention can also be included. We included additional time-constant controls for neighbourhood 'poverty' (i.e. average house price) and 'percentage of non-western minorities'. The latter naturally excludes asylum seekers. At the individual level, we controlled for the time-constant variables gender, age, and education. Our hybrid analysis are based on the total sample of 19,091 respondents living in 2,997 different neighbourhoods.

**Table 1. Descriptive statistics.**

| | Wave 1 | | | | Wave 2 | | | | Δ Wave 2 – Wave 1 | | | |
|---|---|---|---|---|---|---|---|---|---|---|---|---|
| | Mean/% | SD | Min | Max | Mean/% | SD | Min | Max | Mean | SD | Min | Max |
| *Fixed-effects model sample (Nᵢ = 1,389; Nₙᵦ = 1,002)ᵃ* | | | | | | | | | | | | |
| PVV | 24.12 | | | | 75.88 | | | | 51.76 | | | |
| Ethnic threat | 2.92 | 1.14 | 0.00 | 4.00 | 3.08 | 1.00 | 0.00 | 4.00 | 0.16 | 1.12 | -4.00 | 4.00 |
| Interethnic contact | 2.34 | 2.15 | 0.00 | 6.00 | 2.59 | 2.24 | 0.00 | 6.00 | 0.26 | 2.17 | -6.00 | 6.00 |
| Total exposure to asylum seekers | 1.50 | 13.67 | 0.00 | 292.61 | 3.67 | 23.21 | 0.00 | 502.84 | 2.17 | 17.87 | -27.62 | 502.84 |
| Exposure to asylum seekers in regular ASC | 1.40 | 13.42 | 0.00 | 292.61 | 2.01 | 20.36 | 0.00 | 502.84 | 0.61 | 14.12 | -27.62 | 502.84 |
| Exposure to asylum seekers in temporary ASC | 0.10 | 2.64 | 0.00 | 71.38 | 0.66 | 10.26 | 0.00 | 250.29 | 0.56 | 9.81 | 0.00 | 250.29 |
| Exposure to asylum seekers in crisis ASC | 0.00 | 0.00 | 0.00 | 0.00 | 1.00 | 5.19 | 0.00 | 94.06 | 1.00 | 5.19 | 0.00 | 94.06 |
| *Hybrid model sample (Ni = 19,091; Nnb = 2,997)ᵇ* | | | | | | | | | | | | |
| PVVᶜ | 16.72 | | | | 20.48 | | | | 3.76 | | | |
| Ethnic threat | 1.89 | 1.36 | 0.00 | 4.00 | 1.98 | 1.33 | 0.00 | 4.00 | 0.09 | 1.03 | -4.00 | 4.00 |
| Interethnic contact | 2.48 | 2.11 | 0.00 | 6.00 | 2.58 | 2.15 | 0.00 | 6.00 | 0.10 | 1.99 | -6.00 | 6.00 |
| Total exposure to asylum seekers | 1.74 | 15.65 | 0.00 | 690.65 | 3.83 | 23.19 | 0.00 | 689.21 | 2.08 | 16.49 | -31.28 | 502.84 |
| Exposure to asylum seekers in regular ASC | 1.65 | 15.46 | 0.00 | 690.65 | 2.04 | 19.34 | 0.00 | 689.21 | 0.39 | 10.53 | -31.28 | 502.84 |
| Exposure to asylum seekers in temporary ASC | 0.09 | 2.49 | 0.00 | 71.38 | 0.86 | 11.73 | 0.00 | 250.29 | 0.77 | 11.41 | 0.00 | 250.29 |
| Exposure to asylum seekers in crisis ASC | 0.00 | 0.00 | 0.00 | 0.00 | 0.92 | 5.81 | 0.00 | 283.02 | 0.92 | 5.81 | 0.00 | 283.02 |

Sources: 1VOP, COA.

a: 116 respondents living in 81 different neighbourhoods experienced a change in the number of asylum seekers.

b: 1,491 respondents living in 175 different neighbourhoods experienced a change in the number of asylum seekers.

c: In Wave 2 1.75% switched from the PVV to another voting option; 5.51% switched to the PVV from another voting option.

We made a detailed replication package for this paper (including datasets, scripts, additional tests; weblink: https://jochemtolsma.github.io/ExposureASC2020/). It not only allows the interested reader to replicate all our results and claims made in the paper but also to assess the impact of different operationalisations and modelling strategies.

## Results

Among voters who changed their support for the PVV, voters are far more likely to have voted for the PVV in wave 2 compared to wave 1: among the switchers, 75.88% voted for the PVV in wave 2 (Table 1). In general, support for the PVV increased as indicated by the percentages referring to the hybrid sample: from 16.72% in Wave 1 to 20.48% in Wave 2. Although on average the increase in feelings of threat and intergroup contact experiences are relatively small (0.16 and 0.26, respectively for the fixed effects sample), there is quite some within-individual variation in changes in feelings of threat and contact. As expected, we see that asylum seekers are housed in crisis centres only in wave 2. Among our respondents, approximately 8% experienced an inflow of asylum seekers in their neighbourhood. This illustrates that although the 2015 refugee crisis was the biggest refugee crisis the Netherlands experienced in recent history, only a relatively small percentage of voters became directly exposed to asylum seekers in their local neighbourhood environment.

Table 2 displays the results based on the logistic fixed effects models. The positive coefficient for Wave 2 indicates that, irrespective of whether residents have witnessed an influx of asylum seekers in their neighbourhood, the odds of voting for the PVV have increased over time (b = 1.118 se = 0.064, Model 1, Table 2). On top of this general increase in support for the

**Table 2. Fixed effects models predicting voting intention for the PVV (standard errors in parentheses; $N_i$ = 1,389; $N_{nb}$ = 1,002).**

|  | Model 1 | Model 2 | Model 3 | Model 4 | Model 5 |
|---|---|---|---|---|---|
| Wave 2 (wave 1 = ref.) | 1.118* | 1.113* | 1.091* | 1.104* | 1.084* |
|  | (0.064) | (0.064) | (0.064) | (0.064) | (0.065) |
| Exposure to asylum seekers | 0.022+ |  |  |  |  |
|  | (0.012) |  |  |  |  |
| Exposure to asylum seekers in regular ASC |  | 0.008 | 0.008 | 0.008 | 0.008 |
|  |  | (0.014) | (0.015) | (0.014) | (0.014) |
| Exposure to asylum seekers in temporary ASC |  | 0.056 | 0.059 | 0.059 | 0.061 |
|  |  | (0.062) | (0.066) | (0.064) | (0.067) |
| Exposure to asylum seekers in crisis ASC |  | 0.028+ | 0.028+ | 0.029+ | 0.029+ |
|  |  | (0.017) | (0.017) | (0.017) | (0.017) |
| Threat |  |  | 0.268* |  | 0.265* |
|  |  |  | (0.058) |  | (0.059) |
| Contact non-western |  |  |  | 0.039 | 0.034 |
|  |  |  |  | (0.029) | (0.029) |
| Log likelihood | -764.4 | -763.8 | -752.9 | -762.9 | -752.2 |

* $p<0.05$
+ $p<0.10$; (two-tailed test).
Sources: 1VOP, COA.

PVV, people who have suddenly become exposed to asylum seekers as a result of the establishment of an ASC in their neighbourhood are even more likely to switch to the PVV than to switch away from the PVV (b = 0.022, se = 0.012; Model 1, Table 2). This corroborates hypothesis 1. With each unit increase in the exposure to asylum seekers–an increase of one asylum seeker per 1,000 inhabitants–ceteris paribus, the odds for an individual to vote for the PVV increase by 2.2% (exp(0.022)). In Model 2 we break down exposure to asylum seekers by type of ASC. All estimates referring to exposure are positive but only an increase of exposure to asylum seekers housed in crisis centres is significantly related (p<0.10, two-tailed) to people's likelihood to vote for the PVV (b = 0.028, se = 0.017; Model 2, Table 2). That said, the coefficients of all three types of ASCs do not significantly differ from one another: LR chi2(2) = 1.31.

With Model 3, we test the threat mechanism. In line with conflict theory, we find that people who experienced an increase in feelings of ethnic threat are more likely to have switched to the PVV (b = 0.268, se = 0.058; Model 3, Table 2). With one unit increase in ethnic threat, the odds for a single individual to vote for the PVV increase by 31% (exp(0.268)). Increased positive contact with non-western minorities is not significantly related to PVV-voting (Model 4, Table 2) and thus also does not suppress the threat mechanism. Including ethnic threat and contact into our explanatory model simultaneously does not substantially alter the estimates of exposure to asylum seekers (estimates referring to exposure are almost identical across Model 2 and Model 5). Neither increased threat nor contact is more common among residents who experienced increased exposure to asylum seekers than among residents who did not experience an increase in exposure to asylum seekers. There was a significant difference in changes in contact for voters who did not experience an inflow of asylum seekers (M = 0.296, SD = 2.164) and voters who did (M = -0.198, SD = 2.190); t(129.37) = 2.285, p = 0.024 but in the opposite direction as expected. There was no significant difference in changes in threat for voters who did not experience an inflow of asylum seekers (M = 0.170, SD = 1.113) and voters who did (M = 0.108, SD = 1.170); t(127.89) = 0.534, p = 0.594. T-tests performed on fixed-effects sample. Thus, notwithstanding that heightened perceptions of ethnic threat are an

important explanatory factor for changes in PVV support, we therefore refute both hypotheses 2 and 3.

The results of our hybrid models are summarized in Table 3. Results referring to the main variables of interest–exposure to asylum seekers–led to identical conclusions as described above, even though the estimates are smaller than the estimates produced by the fixed effects models, as expected [56]. A change in total exposure to asylum seekers is significantly (p<0.10, two-tailed) related to the odds to vote for the PVV: with each unit increase in the total exposure to asylum seekers the odds to vote for the PVV increases slightly by 0.1% (Model 1). Holding all covariates at their mean, for female voters who did not experience an inflow of asylum seekers the probability to vote for the PVV after the crisis was 13.4%, for female voters who experienced an inflow of 100 asylum seekers per 1,000 neighbourhood residents the probability was 15.1%. For their male counterparts the estimated probabilities are 21.6% and 24.1%, respectively (predicted probabilities based on estimates as summarized in Table 3, Model 1). The impact of increased exposure (change in probabilities of 1.7% for women and 2.5% for men) is substantial in comparison to the general trend in increased support for 'non-treated' voters; increased probabilities of 2.7% for women and 4.0% for men.

The estimates referring to exposure to asylum seekers housed in specific types of ASCs are positive but no longer reach significance (Model 2). When voters' feelings of threat increased (b = 0.117, se = 0.020, Model 3) and, surprisingly, when positive contact with non-western immigrants increased (b = 0.018, se = 0.007, Model 4), their support for PVV increased as well. As can be seen in Model 5, taking into account whether or not people experienced an increase in contact with non-western immigrants did not affect the threat mechanism. Thus our hybrid models also confirm hypothesis 1 and refute hypotheses 2 and 3.

The estimates referring to the time-constant (between) predictors are in line with previous research [14]. Most importantly, voters with higher mean levels of threat are more likely to vote for the PVV (b = 1.414, se = 0.023, Model 3). Voters who have more positive contact with non-western immigrants in their neighbourhood are less likely to vote for the PVV (b = -0.058, se = 0.010, Model 4). The (between-level) estimates referring to exposure to asylum seekers are not significant. This illustrates that the placement of ASCs has been exogenous to people's (pre-crisis) party preference.

## Robustness checks

If people's voting intentions change, they mainly change within the left-wing bloc consisting of Labour (PvdA), GreenLeft (GL) and the Socialist Party (SP) or within the right-wing bloc consisting of liberal-conservatives (VVD), Christian-democrats (CDA) and the PVV [57]. An exception is the exchange between the PVV and the SP, which are both considered to be populist, anti-establishment parties [4,58].

It could be that the PVV lost votes to other parties of the right-wing bloc and won votes from the anti-establishment party SP (or vice-versa), thereby obscuring the general trend of increasing PVV popularity. Moreover, volatility patterns could differ between people who did and did not become exposed to asylum seekers in their neighbourhood in 2015. This may explain in part our small and/or non-significant estimates for our exposure measures reported above. As a robustness check, we therefore also ran multinomial fixed effects models (Table A2 in S1 Appendix; voting intention for the PVV is now the base category). It turns out that during the refugee crisis, the PVV was especially successful in attracting voters from the other anti-establishment party (i.e. SP), as indicated by the estimate of 'wave 2' referring to the odds 'Anti-establishment vs. PVV' (b = -2.008, se = 0.105; this estimate differs significantly from the estimate of 'wave 2' referring to the odds 'Other Parties vs. PVV': LR chi2(1) =

**Table 3. Hybrid models predicting voting intention for the PVV (standard errors in parentheses; $N_i$ = 19,091; $N_{nb}$ = 2,997).**

| | Model 1 | Model 2 | Model 3 | Model 4 | Model 5 |
|---|---|---|---|---|---|
| *Time-varying (within) variables* | | | | | |
| Wave 2 (wave 1 = ref.) | 0.258* | 0.256* | 0.354* | 0.255* | 0.352* |
| | (0.014) | (0.014) | (0.019) | (0.014) | (0.019) |
| Exposure to asylum seekers | 0.001+ | | | | |
| | (0.001) | | | | |
| Exposure to asylum seekers in regular ASC | | 0.001 | 0.002 | 0.001 | 0.002 |
| | | (0.001) | (0.002) | (0.001) | (0.002) |
| Exposure to asylum seekers in temporary ASC | | 0.001 | 0.001 | 0.001 | 0.001 |
| | | (0.001) | (0.001) | (0.001) | (0.001) |
| Exposure to asylum seekers in crisis ASC | | 0.003 | 0.005 | 0.003 | 0.005 |
| | | (0.002) | (0.003) | (0.002) | (0.003) |
| Threat | | | 0.117* | | 0.114* |
| | | | (0.020) | | (0.020) |
| Contact non-western | | | | 0.018* | 0.018+ |
| | | | | (0.007) | (0.010) |
| *Time-constant (between) variables* | | | | | |
| Exposure to asylum seekers | 0.000 | | | | |
| | (0.001) | | | | |
| Exposure to asylum seekers in regular ASC | | -0.000 | 0.001 | -0.000 | 0.000 |
| | | (0.001) | (0.001) | (0.001) | (0.001) |
| Exposure to asylum seekers in temporary ASC | | -0.001 | 0.002 | -0.001 | 0.002 |
| | | (0.003) | (0.003) | (0.003) | (0.003) |
| Exposure to asylum seekers in crisis ASC | | 0.005 | 0.007 | 0.005 | 0.007 |
| | | (0.006) | (0.007) | (0.006) | (0.007) |
| Threat | | | 1.414* | | 1.417* |
| | | | (0.023) | | (0.023) |
| Contact non-western | | | | -0.058* | -0.056* |
| | | | | (0.010) | (0.011) |
| Male (female = ref.) | 0.581* | 0.582* | 0.227* | 0.591* | 0.235* |
| | (0.043) | (0.043) | (0.050) | (0.043) | (0.050) |
| Education | -0.153* | -0.154* | -0.075* | -0.152* | -0.074* |
| | (0.006) | (0.006) | (0.007) | (0.006) | (0.007) |
| Age | -0.014* | -0.014* | -0.012* | -0.015* | -0.013* |
| | (0.002) | (0.002) | (0.002) | (0.002) | (0.002) |
| Proportion non-western minorities neighbourhood | 0.772* | 0.775* | -1.989* | 1.140* | -1.667* |
| | (0.239) | (0.239) | (0.288) | (0.247) | (0.295) |
| Economic deprivation neighbourhood | -0.000 | -0.000 | -0.000 | -0.000 | -0.000 |
| | (0.000) | (0.000) | (0.000) | (0.000) | (0.000) |
| Constant | -2.126* | -2.128* | -5.451* | -1.991* | -5.324* |
| | (0.040) | (0.040) | (0.079) | (0.047) | (0.084) |
| Log likelihood | -17441 | -17440 | -12447 | -17409 | -12426 |

* $p < 0.05$

+ $p < 0.10$; (two-tailed test).

Sources: 1VOP, COA, Statistics Netherlands.

294.87). Of our exposure measures, only 'exposure to asylum seekers in crisis centres' reaches significance and only for the odds 'Right wing vs. PVV' and 'Other parties vs. PVV'. However, all estimates referring to 'exposure to asylum seekers in crisis centres' are negative and do not significantly differ from one another. The impact of exposure to asylum seekers does not depend on pre-crisis voting intentions [59].

Several scholars have suggested that the impact of increasing diversity on related concepts such as prejudice may depend on, for example, residents authoritarian values [60], national rhetoric [61], or the initial share of outgroups [62]. Because this contribution is interested in the average impact of increased exposure to asylum seekers, because even with our impressive dataset, still relatively few neighbourhoods faced an inflow of asylum seekers, and because our survey lacked measures of most likely potential moderators, we did not formulate hypotheses on the conditional impact of exposure to asylum seekers. That said, in additional analyses using our hybrid models, we tested for an interaction between initial levels of ethnic density (i.e. percentage of non-western minorities) and increases in exposure to asylum seekers but these did not reach significance (see replication package).

In line with most previous research on the impact of the ethnic composition of the neighbourhood and support for the radical right, we operationalized exposure to asylum seekers as relative group size, in this case, the number of asylum seekers per 1,000 neighbourhood residents. For crisis centres, our exposure measure refers to the average number of asylum seekers for the days that the centre housed asylum seekers. To try to capture the time heterogeneity in exposure to asylum seekers housed in crisis centres, we multiplied the number of days asylum seekers were housed in each crisis centre (range: 1–41) with the number of asylum seekers at that specific day (range: 20–420). To take into account differences in neighbourhood size, this score was divided by the number of neighbourhood residents (per 1,000). The Spearman's *rho* statistic between the original exposure measure and this alternative operationalization was .53. The alternative operationalization of exposure to asylum seekers in crisis centres did not reach significance in our additional analyses (Table A3 in S1 Appendix). This may indicate that exposure to more asylum seekers is more likely to fuel support for the radical right but that longer exposure to asylum seekers may dampen this effect, possibly due to a familiarization process [55]. In line with this idea, we observe that the estimated impact of the number of days asylum seekers were housed in crisis centres on PVV support was negative (b = -0.113; se = 0.043), and, once we control for differences in time use between crisis centres, our original exposure measure becomes stronger (b = 0.64; se = 0.025; Table A3 in S1 Appendix). Binary exposure measures (yes/no increase in asylum seekers) did not reach significance (Table A3 in S1 Appendix). This operationalization of exposure may simply be too blunt and our non-significant findings may indicate that it is not so much whether asylum seekers were housed in one's neighbourhood but about how many asylum seekers entered the neighbourhood.

As stated above, because people sign up for the 1VOP online panel of their own, there is a self-selection bias in the sample of our respondents. As a robustness check we repeated our fixed effects analysis on a weighted sample (based on sex, age and educational level). For this weighted sample, we find that both exposure to asylum seekers in temporary ASCs and crisis ASCs is related to an increase in support for the PVV (Tables A3 and A4 in S1 Appendix).

We argued above that our study comes close to a natural experiment; voters who experienced an inflow of asylum seekers (the treatment) were similar to voters who did not experience an inflow of asylum seekers. With our individual-level panel data we were already able to take unobserved time-stable heterogeneity into account. With a natural experiment, unobserved time-varying heterogeneity (including pre-treatment trends in PVV support) is not likely to have biased our results. However, we acknowledge that group differences may have occurred by chance and that the distribution of asylum seekers may not have been perfectly

random. As a robustness test, we therefore used a nonparametric pre-processing matching approach on the sample which we used to estimate our fixed effects models. We used a binary treatment variable (increase in exposure = 1) in the matching procedure. The pre-treatment covariates on which we perform a nearest neighbours match are: age, gender, education, threat, contact, ethnic density in neighbourhood and socio-economic status of the neighbourhood. Matching our data leads to a somewhat stronger estimated impact of exposure to asylum seekers in crisis ASCs (Tables A3 and A5 in S1 Appendix).

Following the suggestion of an anonymous reviewer, we also estimated a Difference in Differences estimator. The traditional DiD model for individual-level panel data with additional time-constant covariates ci is:

$$Y_{it} = \beta_1 Time_t + \beta_2 Treat_i + \delta(Time_t \cdot Treat_i) + c_i + \epsilon_{it}, \tag{1}$$

with δ being the DiD estimator and $Treat_i$ the dichotomous treatment variable and $Y_{it}$ the continuous outcome variable. Formula (1) is equivalent to:

$$\Delta Y_i = \beta_1 + \delta\ Treat_i + \epsilon_i \tag{2}$$

Since our outcome is a binary variable there is no standard DiD model. Our main results reported in Table 2 are therefore based on the following model:

$$logit(Pr(\Delta Y_i = 1|Treat_i)) = \beta_1 + \delta\ Treat_i, \tag{3}$$

with $\Delta Y_i$ = 1 if the dependent outcome was 1 post-treatment (i.e. wave 2) and 0 if the dependent variable was 1 pre-treatment (wave 1). Our $Treat_i$ variable is the change in exposure the asylum seekers. Formula (3) is the fixed effects logistic regression model for two waves (or, more precisely, the first difference model which for two waves is equivalent to the more general fixed effects model). Note, that respondents who did not change support for the PVV drop out of this analysis. Because we have a binary outcome and we use a nonlinear link function we cannot interpret δ as the DiD estimator. Moreover, our original 'treatment' variable is not a dichotomous variable, and this also makes why we cannot interpret our effect as the traditional DiD estimator. To be able to interpret our effect as a DiD estimator, we estimated formula (1) directly for a binary outcome variable as an additional robustness check (on our complete sample of panel respondents). That is, we estimated a linear probability model (LPM), while controlling for heteroscedasticity in the error term. We did this once with our original continuous 'treatment' variable and once applying a dichotomization. To avoid any possible post-treatment bias we did not include our time-varying contact or threat measures. We estimated models with and without time-stable covariates. We summarized the DiD estimators in Table A6 (S1 Appendix). The DiD estimators based on binary treatment variables do not reach significance. Above we already observed that it is not whether the neighbourhood experienced an inflow of asylum seekers but how many asylum seekers entered the neighbourhood (Table A3 in S1 Appendix). The DiD estimators based on continuous treatment variables reached significance (albeit only in models without additional covariates) in line with our results reported in Tables 2 and 3.

## Conclusion and discussion

In 2015 an unprecedented number of asylum seekers had to be housed in existing and haphazardly created new (temporary and crisis) asylum seekers centres. Our large-scale individual-level panel data on voting intentions provided us with an unique opportunity to expand academic knowledge about the relationship between the influx of asylum seekers in the local environment and support for the radical right. During the refugee crisis support for the radical

right increased and especially among people who experienced an increase in exposure to asylum seekers in their neighbourhood. In the Netherlands, people appear to be receptive of abrupt, rapid and visible increases in the number of immigrants, which might coincide with a 'not-in-my-back-yard' syndrome [63]. Based on several robustness analyses, we tentatively conclude that larger inflows of asylum seekers in the neighbourhood (relative to the group size of native residents) fuel support for the PVV, at least in the short run. When asylum seekers stay in the neighbourhood for longer, this impact may be curbed [55]. We encourage scholars to replicate our findings, preferably in different countries.

We tested our hypotheses employing individual-level panel data allowing us to control for (time-stable) unobserved heterogeneity. Given the short time-window between our survey waves, selective residential mobility did not plague our study. Moreover, exposure to asylum seekers was to a large extent random and our study therefore resembled to some extent a natural experiment. Because of these three reasons, combined with the fairly consistent results over different modelling strategies, many data and model requirements are met to give a causal interpretation to our finding that an inflow of asylum seekers into the neighbourhood is related to an increase in radical right support in this neighbourhood. However, we need to acknowledge that a natural experiment is not a true experiment and that our estimates only reached the boundary of the conventional significance criteria.

War refugees are considered to be more deserving to enter and stay in the Netherlands than 'classical migrants' [15]. Consequently, the impact of similar unexpected sharp increases in the size of migrant populations in neighbourhoods that result from union formation and labour migration on radical right voting may be larger. Naturally, these latter migration flows are generally less volatile than migration flows as a result of a humanitarian crisis and this may explain why previous research only observed small effects of changing migrant sizes in neighbourhoods on radical right voting [14].

This study is one of the first to demonstrate the previously established threat-radical right relationship [31,32] from a longitudinal perspective; residents whose worries about neighbourhood deterioration resulting from migration increased during the refugee crisis were more likely to start expressing intentions to vote for the PVV. Even though our results thus confirmed that increasing feelings of local ethnic threat are an important driving force for support for the radical right, this could not explain why especially residents of neighbourhoods in which asylum seekers were housed became more likely to vote for the PVV. This is because an inflow of asylum seekers did not increase feelings of neighbourhood deterioration. We need to acknowledge that our single-item threat measure did not explicitly refer to economic, cultural or safety issues in the neighbourhood because of the inflow of asylum seekers. With a better (multi-item) measurement instrument we may have picked up the assumed relation between the inflow of asylum seekers and increased feelings of ethnic threat. However, our null finding, is in line with previous studies, using different measures of ethnic threat.

It could be the case that not necessarily anti-immigrant attitudes increased as a result of the influx of asylum seekers but that already previously held (negative) opinions regarding immigration became more salient in these neighbourhoods. This resonates with the idea of Karreth *et al.* [59] that increasing diversity is only related with negative attitudes towards immigrants among people on the political right. At the national level, the share of the Dutch population that mentioned immigration as one of the two most important issues facing the Netherlands at the moment increased between February 2015 and November 2015 from 9% to 56% [64,65]. Heightened issue salience is likely to increase the relative importance that voters attach to this issue. A promising direction for future research would therefore be to assess the role of (increased) issue saliency in the link between (increased) local outgroup size and support for the radical right.

We expected that increased positive contact with minorities would mitigate the impact of increased exposure to asylum seekers on radical right voting. In line with contact theory, we observed that voters who have more contact with non-western minorities are less likely to vote for the radical right as compared to voters with less contact with non-western minorities. Unexpectedly, at the same time we observed that residents who experienced an *increase* in positive interethnic contact during the refugee crisis became more likely to express support for the radical right; the estimates referring to the between and within effects of interethnic contact were opposite in direction. Increased positive interethnic contact did not suppress the threat mechanism. As contact opportunities may lead to both positive and negative contact, it may be that the same voters who experienced an increase in positive contact also experienced an increase in negative contact and that the impact of negative contact experiences on voting intentions was more severe. A recent study showed that negative contact experiences predict support for the PVV, even after controlling for indicators of threat [50].

Political parties had to come clean during the refugee crisis as to their position on the immigration issue in general and as to their position on whether and where to house asylum seekers in particular. The VVD (next to the PVV the main anti-immigration party) definitely showed colours by making it perfectly clear that to limit the influx of even more asylum seekers it was willing to endorse the EU-Turkey agreement (i.e. the 'refugee deal'). This may explain why the PVV was not especially successful in attracting new voters from the VVD during the refugee crisis. Instead, we demonstrated that especially former voters for the Social Party (SP) were likely to switch to the PVV. Volatility patterns are thus clearly context dependent. The SP shares the radical right's anti-establishment rhetoric, but as owner of the issue of immigration [66] the radical right used its anti-establishment rhetoric to successfully capitalize on the convergence of the immigration and anti-establishment issue during the 2015 refugee crisis at the expense of the Socialist Party.

The radical right gained support in the Netherlands during the 2015 refugee crisis and especially among residents who were exposed to asylum seekers 'in their backyard'. Why this is so remains unclear; the threat mechanism does not seem to hold. We encourage scholars to test two alternative ideas we could unfortunately not test ourselves: the negative contact mechanism and the issue salience argument.

## Supporting information

**S1 Data.**
(DTA)

**S1 Appendix.**
(DOCX)

## Author Contributions

**Conceptualization:** Jochem Tolsma, Michael Savelkoul.

**Data curation:** Jochem Tolsma.

**Formal analysis:** Jochem Tolsma, Joran Laméris.

**Methodology:** Jochem Tolsma.

**Supervision:** Jochem Tolsma.

**Writing – original draft:** Joran Laméris.

**Writing – review & editing:** Jochem Tolsma, Michael Savelkoul.

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
