## [Decision Letter · Decision Letter 0]

21 Oct 2020

PONE-D-20-21671

Exposure to Asylum Seekers and Changing Support for the Radical Right: A Natural Experiment in the Netherlands

PLOS ONE

Dear Dr. Tolsma,

Thank you for submitting your manuscript to PLOS ONE. After careful consideration, we feel that it has merit but does not fully meet PLOS ONE’s publication criteria as it currently stands. Therefore, we invite you to submit a revised version of the manuscript that addresses the points raised during the review process.

While revising the manuscript, I kindly ask you to focus on the issues raised by Reviewer 2, who recommended to reject it. Upon reception of the revised manuscript, I am going to send it along to Reviewer 1 (who asked to revise a few minor points) and Reviewer 2 for the second round. 

We look forward to receiving your revised manuscript.

Kind regards,

Shang E. Ha, Ph.D.

Academic Editor

PLOS ONE

Journal Requirements:

2.We note that you have indicated that data from this study are available upon request. PLOS only allows data to be available upon request if there are legal or ethical restrictions on sharing data publicly. For more information on unacceptable data access restrictions, please see http://journals.plos.org/plosone/s/data-availability#loc-unacceptable-data-access-restrictions.

Reviewers' comments:

Reviewer's Responses to Questions

**Comments to the Author**

1. Is the manuscript technically sound, and do the data support the conclusions?

Reviewer #1: Yes

Reviewer #2: Partly

Reviewer #3: Yes

2. Has the statistical analysis been performed appropriately and rigorously? 

Reviewer #1: Yes

Reviewer #2: No

Reviewer #3: Yes

3. Have the authors made all data underlying the findings in their manuscript fully available?

Reviewer #1: Yes

Reviewer #2: Yes

Reviewer #3: Yes

4. Is the manuscript presented in an intelligible fashion and written in standard English?

Reviewer #1: Yes

Reviewer #2: Yes

Reviewer #3: Yes

5. Review Comments to the Author

Reviewer #1: Title: Exposure to Asylum Seekers and Changing Support for the Radical Right

This paper tests how the sudden inflow of asylum seekers to a region changes voters’ support for the radical right in the region. Unlike many other studies on the support for the radical right, this paper utilizes an individual-level panel dataset on the regional level, employs a natural experiment method, and distinguishes between different types of refugee centers. Though authors’ finding that the inflow of asylum seekers strengthens the support for the radical right is not surprising, their methodological approaches contribute to the literature. Though I support the publication of this manuscript, there are some questions that should be answered beforehand.

1. The structure of the hypotheses is odd. Hypothesis 1 describes the positive correlation between refugee inflow and support for the PVV. Then, the two hypotheses (2a and 2b) suggest two different causal mechanisms between the two: one for the positive correlation (threat) and the other for the negative correlation (contact). I do not deny that the inflow of asylum seekers, or immigrants in general, can have dual effects (both positive and negative) through threat and contact mechanisms either on the support for the radical right or on public opinion on immigrants. Nonetheless, I think authors need to re-frame their hypotheses so that they incorporate all of these possible mechanisms and patterns.

2. Related to the first comment, authors find and conclude that their results support H1, but not H2a and H2b. That is, though they find a positive correlation between refugee inflow and PVV support, the causal mechanisms they hypothesized are not supported by the results. Then, the question is: WHY does the refugee inflow strengthen support for the radical right? Authors need to discuss this.

3. Though authors distinguish economic threat and cultural threat in their theory section, their threat variable doesn’t. I understand that authors were not able to change the survey question, but they need to provide more discussion on the question wording itself. When the question was delivered to respondents, did it imply economic threat, cultural threat, or both of them to the people?

4. As authors acknowledge, one critical weakness of their data is a plausible self-selection bias because respondents basically voluntarily sign up for the survey. One question related to this is: is there possibility that PVV supporters, after the sudden inflow of asylum seekers, are more motivated to accept the invitation to the survey in the 2nd wave because of, for example, their anger from the inflow? If this is true, then the self-selection bias problem occurred and it could make their results biased toward their findings. So, is there any way to make sure that PVV supporters and non-supporters had the same propensity to sign up for the survey, both in the 1st and the 2nd wave?

5. Authors describe three different types of refugee centers on page 11, but I don’t believe that they explained what a temporary ASC is. (Does the “crisis ASCs” on line 320 actually mean temporary ASCs?)

Reviewer #2: This is an interesting paper on how the sudden inflow of refugees influenced vote intentions in the Netherlands. The strength of the paper is the possibly exogenous exposure to refugees due to the rapid inflow, combined with individual level panel data. The authors find that exposure to refugees increased the vote intention for PVV, the anti-immigration party in the study.

My main issue with the paper is how they analyze the data. The current analysis makes me not convinced that the authors estimate the effects of exposure.

The first issue regards exogeneity. The authors have a clear ambition to estimate causal effects, but is not sufficiently clear on what variation in exposure to refugees that is exogenous. Clearly, between-unit variation in exposure is not exogenous, but part of the variation between the waves might be exogenous. To me, the most promising source of exogenous variation stems from the crisis ASC, and the paper and the analysis should be centered on that source of variation.

This leads me to the second issue which regards the analysis. I think the authors should estimate a standard differences-in-differences model using the crisis ASC as the treatment indicator (equal to one if a crisis ASC was set up in the neighbourhood between wave 1 and 2) and the wave as the post-treatment indicator. The DD estimate from this analysis might be given a causal interpretation.

This again leads me to the third issue which is the examination of “as-if-random” exposure to refugees. The balance analysis in the paper (Table A3) is not properly explained. The appropriate way to examine balance is to conduct an F-test of whether the exogenous covariates can jointly predict the treatment. For instance, if you use the setup of crisis ASC in the respondents as the treatment you need to show that treated neighborhoods are similar to comparison neighborhoods.

The fourth issue is how the authors threat the contact and threat variables. To me these variables should be analyzed as outcomes that are potentially affected by exposure, they should not be analyzed as covariates (see the literature on post-treatment bias). I understand that the authors considers them as mechanisms or mediators, but one needs separate exogenous variation to properly estimate the role of mediators. Or, if the authors are willing to make strong (and in my view implausible) assumptions, they conduct a Baron-Kenny-type of mediation analysis.

Other issues. i) The attrition analysis mentioned on page 9 should be reported in the appendix. ii) The underrepresentation of different groups (pg 9) should be presented in the appendix. Also, are sample weights applied? iii) The authors are probably interested in the experimental literature on contact theory, reviewed in Paluck et al. (2019, The contact hypothesis re-evaluated, Behavioural Public Policy) iv) Have they considered non-linearity in the relationship between exposure and vote intention, perhaps exposure has larger effects in neighborhoods with low prior exposure (e.g. Hopkins 2010, cited in the paper)?

Reviewer #3: This is a very important study. The effect to contact with asylum seekers on support for radical right-wing populist parties is an important subject to study. This is not the first study to examine this, but it is a very rigorous study. In this field, in particular given the social relevance of this research (how do values like tolerance develop, what is the societal reaction to refugees) and academic relevance in the debate between contact and threat, I think that studies like these, which rigorously look at the effect of specific events are welcome.

In particular I think this contributions stands out because of their advanced quasi-experimental design, which is apt to study the phenomenon, well-executed and convincing.

I have only very minor concerns. The first of these is that there is a typo on p.13: it now reads concerning the main effect studied in the paper “(b=0.022, se=0.12; Model 1, Table 2)” but then the effect would not be significant. This has to be (b=0.022, se=0.012) in line with the Table 2.

The second is that the paper mixes the terms refugees and asylum seekers while the first term refers to people who have an official status as refugee and these second term refers to people who want that status. That means that in COA centres there only are asylum seekers and no refugees (who get their own housing once their status has been given).

6. PLOS authors have the option to publish the peer review history of their article (what does this mean?). If published, this will include your full peer review and any attached files.

Reviewer #1: No

Reviewer #2: No

Reviewer #3: No

---

## [Author Response · Author response to Decision Letter 0]

1 Dec 2020

Dear prof. Shang, E. Ha, 

I kindly refer to the rebuttal letter and the revised cover letter. 

yours sincerely, 

Jochem

---

## [Decision Letter · Decision Letter 1]

6 Jan 2021

Exposure to Asylum Seekers and Changing Support for the Radical Right: A Natural Experiment in the Netherlands

PONE-D-20-21671R1

Dear Dr. Tolsma,

We’re pleased to inform you that your manuscript has been judged scientifically suitable for publication and will be formally accepted for publication once it meets all outstanding technical requirements. (And we recommend you to consider incorporating some of the remaining issues Reviewer 2 raised in the final version of your manuscript.) 

Kind regards,

Shang E. Ha, Ph.D.

Academic Editor

PLOS ONE

Additional Editor Comments (optional):

Reviewers' comments:

Reviewer's Responses to Questions

**Comments to the Author**

1. If the authors have adequately addressed your comments raised in a previous round of review and you feel that this manuscript is now acceptable for publication, you may indicate that here to bypass the “Comments to the Author” section, enter your conflict of interest statement in the “Confidential to Editor” section, and submit your "Accept" recommendation.

Reviewer #1: All comments have been addressed

Reviewer #2: (No Response)

2. Is the manuscript technically sound, and do the data support the conclusions?

Reviewer #1: Yes

Reviewer #2: Partly

3. Has the statistical analysis been performed appropriately and rigorously? 

Reviewer #1: Yes

Reviewer #2: Yes

4. Have the authors made all data underlying the findings in their manuscript fully available?

Reviewer #1: Yes

Reviewer #2: Yes

5. Is the manuscript presented in an intelligible fashion and written in standard English?

Reviewer #1: Yes

Reviewer #2: Yes

6. Review Comments to the Author

Reviewer #1: Authors tried to answer for all of my questions as much as they could. So, I'm happy to recommend the publication of this manuscript.

Reviewer #2: The authors have done a good job revising the paper. In particular I appreciate the DiD-analysis presented in Table A6. To me this is the most straightforward and convincing test of the claims in the paper. Therefore I think it should be included in the main paper, also because the conclusions modify the impression from the discussion of the results from the authors' main analysis: The estimates in Table A6 seem to be very small (and model dependent) and not politically important. I think the paper should make that clear, so that the readers do not believe that the effects are more important than they are. Also, standard errors in Table A6 should be adjusted for clustering on neighbourhood, it is not clear whether that is currently the case.

7. PLOS authors have the option to publish the peer review history of their article (what does this mean?). If published, this will include your full peer review and any attached files.

Reviewer #1: No

Reviewer #2: No

---

## [Editor Report · Acceptance letter]

20 Jan 2021

PONE-D-20-21671R1 

Exposure to Asylum Seekers and Changing Support for the Radical Right: A Natural Experiment in the Netherlands 

Dear Dr. Tolsma:

I'm pleased to inform you that your manuscript has been deemed suitable for publication in PLOS ONE. Congratulations! Your manuscript is now with our production department. 

Kind regards, 

on behalf of

Dr. Shang E. Ha 

Academic Editor

PLOS ONE